# EvalAlign: Supervised Fine-Tuning Multimodal LLMs with Human-Aligned Data for Evaluating Text-to-Image Models

## Abstract

The recent advancements in text-to-image generative models have been remarkable. Yet, the field suffers from a lack of evaluation metrics that accurately reflect the performance of these models, particularly lacking fine-grained metrics that can guide the optimization of the models. In this paper, we propose EvalAlign, a metric characterized by its accuracy, stability, and fine granularity. Our approach leverages the capabilities of Multimodal Large Language Models (MLLMs) pre-trained on extensive data. We develop evaluation protocols that focus on two key dimensions: image faithfulness and text-image alignment. Each protocol comprises a set of detailed, fine-grained instructions linked to specific scoring options, enabling precise manual scoring of the generated images. We supervised fine-tune (SFT) the MLLM to align with human evaluative judgments, resulting in a robust evaluation model. Our evaluation across 24 text-to-image generation models demonstrate that EvalAlign not only provides superior metric stability but also aligns more closely with human preferences than existing metrics, confirming its effectiveness and utility in model assessment. We will make the code, data, and pre-trained models publicly available.

## 1 Introduction

Text-to-image models, such as DALL·E series (Ramesh et al., 2022; Betker et al., 2023), Imagen (Saharia et al., 2022), and Stable Diffusion (Podell et al., 2023), have significantly impacted various domains such as entertainment, design, and education, by enabling high-quality image generation. These technologies not only advance the field of text-to-image generation but also bloom applications such as video generation (Blattmann et al., 2023; Zhang et al., 2023d; Tan et al., 2024b), image editing (Song et al., 2021; Huang et al., 2023b; Zhang et al., 2023c) and human image generation (Wang et al., 2024). Despite achieving incredible progress, the evaluation methods in this area are far from flawless and suffer heavily from data bias, as they are mainly trained on real images but are employed to evaluate synthesized images.

Since human-based evaluations are considerably costly in money and time, existing evaluation methods are primarily based on pretrained models, which are trained on real images. However, the trained real images are generated by humans and high in image faithfulness and text-image alignment because of their generation essence. Meanwhile, the evaluated images are synthesized by text-to-image models and encounter problems such as low image faithfulness or text-image alignment, constrained by the performance of generative models.

We dub the gap between the training data and the evaluated data as data bias, which may cause the evaluation models perform ill-suited on text-to-image evaluation. Because of the data bias, existing text-to-image evaluation methods performs poorly in synthesized image evaluations. Unfortunately, during our preliminary observation, nearly every synthesized images contain visual elements with low image faithfulness or text-image alignment, emphasize their significance on evaluation performance. Notably, there are also some works such as HPSv2 (Wu et al., 2023b) and PickScore Kirstain et al. (2024), where their evaluation models are trained synthesized images. However, in their evaluation settings, the utilized synthesized images are treated as real images as they don't explicitly recognize the problem of synthesized images with low image faithfulness.

In view of these issues, we propose EVALALIGN, a comprehensive, fine-grained and interpretable metric on text-to-image model assessing with low cost but high accuracy. To build EVALALIGN, we first curate a dataset composed of fine-grained human feedback scores on synthesized images, with consideration of the corresponding prompts. The granularity of the feedback covers 11 skills categorized into two aspects: image faithfulness and text-image alignment. After that, we Supervised finetune (SFT) a Multimodal Large Language Model (MLLM) on the annotated dataset, aligning it with human prior on detailed and accurate text-to-image evaluation.

Owing to extensive pre-training and large model capacity, MLLMs demonstrate excellent image-text understanding and generalization capabilities. However, since the pre-training data does not include synthesized images with low image faithfulness or evaluation-related text instructions, using MLLMs directly for model evaluation may yield non-optimal results. Especially, we want to use MLLMs to support comprehensive and detailed evaluations, encompassing 11 skills and 2 aspects. The definitions and nuances of these may not be fully understood by the MLLM. Therefore, we employ SFT on a small amount of high-quality annotated data to align the MLLM with human judgement on evaluating synthesized images in criteria of 11 skills and 2 aspects. Notably, since the main intelligence of EVALALIGN stems from the annotated dataset and the utilized MLLM, we will make them accessible to the public.

In summary, our main contributions can be summarized as follows:

- We build a detailed human feedback dataset specifically designed to address the aforementioned challenges of text-to-image model evaluations. The annotated dataset is thoroughly cleaned, carefully balanced in topics, and systematically annotated by human. The dataset is composed by fine-grained human prior on evaluating synthesized images in criteria of 11 skills and 2 aspects.
- We propose EVALALIGN, a text-to-image evaluation method which accurately aligns evaluation models with fine-grained human prior using the annotated dataset. EVALALIGN exclusively supports an accurate, comprehensive, fine-grained and interpretable text-to-image evaluations. Besides EVALALIGN is cost-effective in terms of annotation and training and computationally efficient.
- With EVALALIGN, we conduct evaluations over 24 text-to-image models and compare EVALALIGN with existing evaluation methods. Quantitative and qualitative experiments demonstrate that EVALALIGN outperforms other methods in evaluating model performance.

## 2 RELATED WORK

### 2.1 BENCHMARKS OF TEXT-TO-IMAGE GENERATION

Despite the incredible progress achieved by text-to-image generation Zhang et al. (2023a); Tan et al. (2024a), evaluations and benchmarks in this area are far from flawless and contain critical limitations. For example, the most commonly used metrics, IS (Salimans et al., 2016), FID (Heusel et al., 2017), and CLIPScore (Hessel et al., 2021) are broadly recognized as inaccurate for their inconsistency with human perception. To address, HPS series (Wu et al., 2023b;a), PickScore (Kirstain et al., 2024), and ImageReward (Xu et al., 2024) introduced human preference prior on image assessing to the benchmark, thereby allowing better correlation with image quality. However, with varying source and size of training data, these methods merely score the evaluated images in a coarse and general way, which cannot serve as an indication for model evolution. Meanwhile, HEIM (Lee et al., 2024) combined automatic and human evaluation and holistically evaluated text-to-image generation in 12 aspects, such as alignment, toxicity, and so on. As a consequence, HEIM relies heavily on human labour, limiting its application within budget-limited research groups severely. Otani et al. (2023) standardized the protocol and settings of human evaluation, ensuring its verifiable and reproducible. Considering the issues of existing benchmarks, we propose EVALALIGN to offer a cost-efficient, comprehensive and fine-grained text-to-image model evaluation. Through our observations, we found that image faithfulness and text-image alignment are two key factors for comprehensive evaluation. Image faithfulness requires the model to generate visual elements that are consistently faithful to the real-world. For example, visual elements such as distorted body. Meanwhile, text-image alignment measures how the generated images are aligned with their corresponding prompts.

There are also some works bear a resemble with us. For instance, TIFA (Hu et al., 2023), Gecko (Wiles et al., 2024) and LLMScore (Lu et al., 2024) also formulate the evaluation as a set of visual question

Table 1: **Comparison of different evaluation metrics and frameworks for text-to-image generation.** EVALALIGN focuses on two key evaluation aspects, i.e., image faithfulness and text-image alignment, and supports human-aligned, fine-grained, and automatic evaluations. P: Prompt. I: Image. A: Annotation.

| Method | Venue | Benchmark Feature | | | Dataset Size | | | Evaluation Aspect | |
|---|---|---|---|---|---|---|---|---|---|
| | | Human-aligned | Fine-grained | Automatic | P | I | A | Faithfulness | Alignment |
| Inception Score (Salimans et al., 2016) | NeurIPS 2016 | ✗ | ✗ | ✓ | – | 1.3M | – | ✓ | ✗ |
| FID (Heusel et al., 2017) | NeurIPS 2017 | ✗ | ✗ | ✓ | – | 1.3M | – | ✓ | ✗ |
| CLIP-score (Hessel et al., 2021) | EMNLP 2021 | ✗ | ✗ | ✓ | 400M | 400M | – | ✗ | ✓ |
| HPS (Wu et al., 2023b) | ICCV 2023 | ✓ | ✗ | ✓ | 25K | 98K | 25K | – | – |
| TIFA (Hu et al., 2023) | ICCV 2023 | ✓ | ✓ | ✓ | 4K | – | 25K | ✗ | ✓ |
| TVRHE (Otani et al., 2023) | CVPR 2023 | ✓ | ✗ | ✗ | – | – | – | ✓ | ✗ |
| ImageReward (Xu et al., 2024) | NeurIPS 2023 | ✓ | ✗ | ✓ | 8.8K | 68K | 137K | – | – |
| PickScore (Kirstain et al., 2024) | NeurIPS 2023 | ✓ | ✗ | ✓ | 35K | 1M | 500K | – | – |
| HPS v2 (Wu et al., 2023a) | arXiv 2023 | ✓ | ✗ | ✓ | 107K | 430K | 645K | – | – |
| HEIM (Lee et al., 2024) | NeurIPS 2023 | ✓ | ✓ | ✗ | – | – | – | ✓ | ✓ |
| Gecko (Wiles et al., 2024) | arXiv 2024 | ✓ | ✓ | ✓ | 2K | – | 108K | ✗ | ✓ |
| LLMScore (Lu et al., 2024) | arXiv 2024 | ✓ | ✓ | ✓ | – | – | – | ✗ | ✓ |
| EVALALIGN (ours) | – | ✓ | ✓ | ✓ | 3K | 21K | 132K | ✓ | ✓ |

answering procedure and use LLMs as evaluation models. However, while they all mainly focus on text-image alignment, our approach takes both text-image alignment and image faithfulness into consideration. Moreover, the evaluation of LLMScore requires an object detection stage, which introduces significantly extra inference latency to the evaluation pipeline.

As illustrated in Table 1, existing text-to-image evaluation methods contains various limitations, making them incapable to serve as a fine-grained, comprehensive, and human-preference aligned automatic benchmark. While our work fills in this gap economically, and can be employed to indicate evolution direction and support thorough analysis of text-to-image generation models.

## 2.2 MULTIMODAL LARGE LANGUAGE MODELS (MLLMs)

Pre-trained on massive text-only and image-text data, MLLMs have exhibited exceptional image-text joint understanding and generalization abilities, facilitating a large spectrum of downstream applications. Among the works major in MLLMs, LLaVA (Liu et al., 2024b; 2023) and MiniGPT4 (Zhu et al., 2023; Chen et al., 2023a) observed that multimodal SFT is sufficient to align MLLMs with human preferences and enable them to accurately answer fine-grained questions about visual content. Besides, Video-LLaMA (Zhang et al., 2023b) and VideoChat (Li et al., 2023) utilized MLLMs for video understanding. VILA (Lin et al., 2023) quantitatively proved that involving text-only instruction-tuning data during SFT can further ameliorate model performance on text-only and multimodal downstream tasks. LLaVA-NeXT (Liu et al., 2024a) extracted visual tokens for both the resized input image and the segmented sub-images to provide more detailed visual information for MLLMs, achieving significant performance bonus on tasks with high-resolution input images.

However, due to the data bias, existing MLLMs cannot perfectly quantify for text-to-image evaluations. Thus, we meticulously curate a SFT dataset to align MLLMs with detailed human feedback on synthesized images.

## 3 EVALALIGN DATASET CONSTRUCTION

To train, validate and test the effectiveness of our evaluation models, we build EVALALIGN dataset. Specifically, EVALALIGN dataset is a meticulously annotated collection featuring fine-grained annotations for images generated on text conditions. This dataset comprises 21k images, each accompanied by detailed instructions. The compilation process for the EVALALIGN Dataset encompasses prompt collection, image generation, and precise instruction-based annotation.

### 3.1 PROMPTS AND IMAGES COLLECTION

**Prompt collection.** To assess the capabilities of our model in terms of image faithfulness and text-image alignment, we collect, filter, and clean prompts from existing evaluation datasets and generated prompts based on LLM. These prompts encompass a diverse range from real-world user prompts, prompts generated through rule-based templates with LLM, to manually crafted prompts. Specifically,

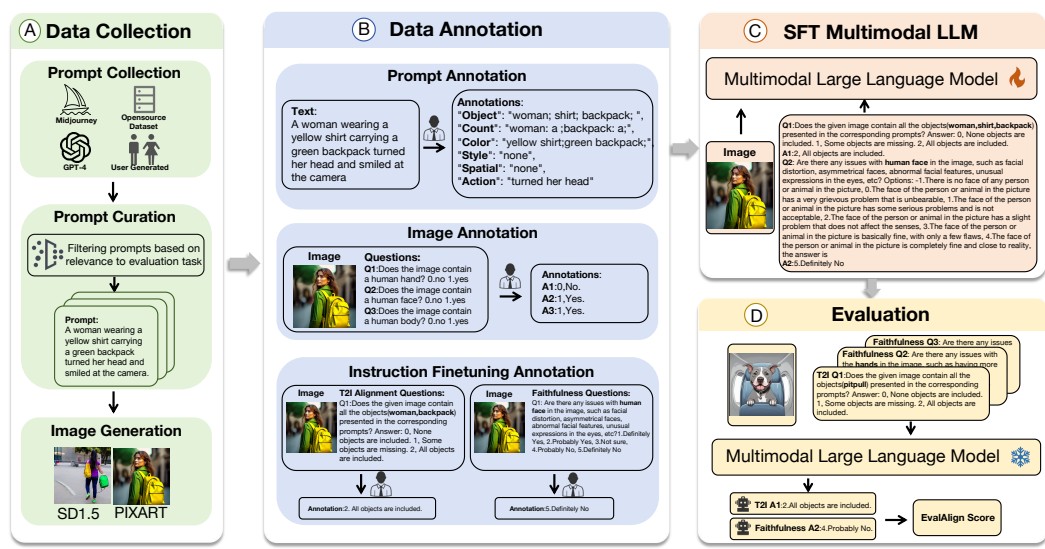

Figure 1: **Overview of EVALALIGN.** We collect, filter and clean prompts from various sources to ensure their quantity, quality and diversity. We use 8 state-of-the-art text-to-image models to the generate images for evaluation. These synthesized images are then delegated to human annotators for thorough multi-turn annotation. Finally, the annotated data are used to finetune a MLLM to align it with fine-grained human preference, thereby adapting the model to perform text-to-image evaluation on image faithfulness and text-image alignment.

the utilized prompts are sourced from HPS (Wu et al., 2023b), HRS-Bench (Bakr et al., 2023), HPSv2 (Wu et al., 2023a), TIFA (Hu et al., 2023), DSG (Cho et al., 2023a), T2I-Comp (Huang et al., 2023a), Winoground (Thrush et al., 2022), DALL-EVAL (Cho et al., 2023b), DiffusionDB (Wang et al., 2023), PartiPrompts (Yu et al., 2022), DrawBench (Saharia et al., 2022), and JourneyDB (Sun et al., 2024).

**Prompt curation.** To facilitate a clean and reasonable evaluation, each prompt to be annotated have to instruct text-to-image models to generate images that can reflect model performances on image faithfulness and text-image alignment. However, considering some of the collected prompts fail to achieve the purpose, we need to filter and balance the collected prompts to ensure their quantity, quality and diversity. For image faithfulness evaluation, we prioritize prompts related to human, animals, and other tangible objects, as prompts depicting sci-fi scenarios are less suitable for this type of assessment. Consequently, the prompt filter for image faithfulness initially selects prompts that describe human, animals, and other real objects. After deduplicating these prompts, we carefully select 1,500 distinct prompts with varying topic, background and style. The selected prompts encompass 10k subjects across 15 categories. For text-image alignment evaluation, we refine our selection based on descriptions of style, color, quantity, and spatial relationships in the prompts. Specifically, only prompts contain relevant descriptions and exceed 15 words in length are considered, culminating in a final set of 1,500 prompts.

**Image generation.** To train and evaluate the MLLM, we use a diverse set of images generated by various models using the aforementioned prompts, facilitating detailed human annotation. For each prompt, multiple images are generated across different models. The models used to generate these images vary in architectures and scales, enhancing the dataset diversity. There are 24 models used to generate these images, varying in architecture as well as scale and thus enhancing the dataset diversity. For detailed information on the generation setting of each model, please refer to the appendix.

The training and validation set comprises synthesized images from 8 out of the 24 models, whereas the test set spans all of them. Particularly, the exclusive inclusion of the 16 models in the test set is crucial for validating the MLLM's ability to generalize beyond its training data. Through our manual inspection, in this way, we attain ample synthesized images with a balanced diversity in the performance of image faithfulness and text-image alignment.

## 3.2 Data Annotation

**Prompt annotation.** For text prompts focused on text-image alignment, we begin by annotating the entities and their attributes within the text, as illustrated in Figure 1. Our annotators extract the entities mentioned in the prompts and label each entity with corresponding attributes, including quantity, color, spatial relationships, and actions. During the annotation, we also ask the annotators to annotate the overall style of the image if described in the corresponding prompt and report prompts that contain toxic and NSFW content. These high-quality and detailed annotations facilitate the subsequent SFT training and evaluation of the MLLM. The prompt annotation procedure ensures that the MLLM can accurately align and respond to the nuanced details specified in the prompts, enhancing both the training process and the model's performance in generating images that faithfully reflect the described attributes and style.

**Image annotation.** The images generated by text-to-image models often present challenges such as occluded human body parts, which can impede the effectiveness of SFT training and evaluation of the MLLM. To address these challenges and enhance the model's training and evaluative capabilities, specific annotations are applied to all images depicting human and animals. These annotations include: presence of human or animal faces; visibility of hands; visibility of limbs. By implementing these annotations, we ensure that the MLLM can more effectively learn from and assess the completeness and faithfulness of the generated images. This structured approach to annotation not only aids in identifying common generation errors but also optimizes the model's ability to generate more accurate and realistic images, thereby improving both training outcomes and the model's overall performance in generating coherent and contextually appropriate visual content.

**Instruction-finetuning data annotation.** To align the MLLM with human preference prior on detailed synthesized image assessing, we can train the model on a minimal amount of fine-grained human feedback data through SFT training. As a consequence, we devise two sets of questions, each is concentrated on a specific fine-grained skill of image faithfulness and image-text alignment. Human annotators are required to answer these questions to acquire the fine-grained human preference data. To aid them to understand the meaning and principle of each question, thereby ensuring high annotation quality, we employ a thorough and comprehensive procedure of annotation preparation. First, we write a detailed annotation guideline and conduct a training for the annotators to explain the annotation guideline and answer their questions about the annotation. Then, we conduct a multi-turn trial annotation on another 50 synthesized images. After each trial, we calculate the Cohen's kappa coefficient and interpret annotation guidelines for our annotators. In total, we conduct nine turns of trial annotation, and in the last turn of the trial, the Cohen's kappa coefficient of our annotators reaches 0.681, indicating high inter-annotator reliability and high annotation quality.

After completing the aforementioned preparations, we delegate the images filtered during image annotation to 10 annotators and ask them to complete the annotation just as how they did in the trial annotation. Furthermore, during the whole annotation procedure, four experts in text-to-image generation conduct random sampling quality inspection on the present annotated results, causing a second and a third re-annotation on 423 and 112 inspection-failed samples. Overall, owing to the valuable work of our human annotators and our fastidious annotation procedure, we get quality-sufficient instruction-tuning data required for the SFT training of the MLLM. More details of the annotation procedure will be introduced in supplementary files.

## 3.3 Dataset Statistics

To summarize, we generate 24k images from 3k prompts based on 8 text-to-image models, which includes DeepFloyd IF (Alex Shonenkov & et al., 2023), SD15 (Rombach et al., 2022), LCM (Luo et al., 2023), SD21 (Rombach et al., 2022), SDXL (Podell et al., 2023), Wuerstchen (Pernias et al., 2023), Pixart (Chen et al., 2023b), and SDXL-Turbo (Stability AI, a). After data filtering, 4.5k images are selected as annotation data for task of text-image alignment. Subsequently, these images are carefully annotated to generate 13.5k text-image pairs, where 11.4k are used to the training dataset and 2.1k to the validation dataset. For the image faithfulness task, we select 12k images for annotation, yielding 36k text-image pairs, with 30k are used to the training dataset and 6.2k to the validation dataset. Additionally, we employed 24 text-to-image models to generate 2.4k images from 100 prompts. After annotation, these images are used as testing dataset. Figure 2 and Figure 3 show

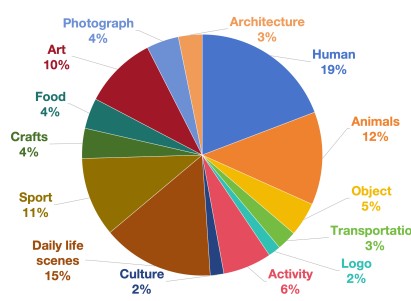 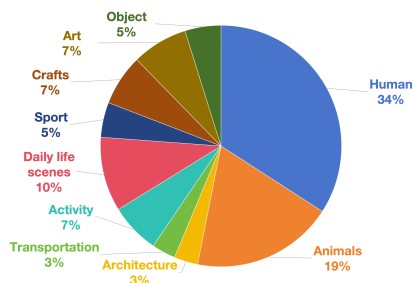

Figure 2: **Statistics of prompts on evaluating text-to-image alignment.** Prompts in our text-to-image alignment benchmark covers a broad range of concepts commonly used in text-to-image generation.

Figure 3: **Statistics of prompts on evaluating image faithfulness.** Prompts in our image faithfulness benchmark covers a broad range of objects and categories that related to image faithfulnes.

the distribution of objects in different categories within our prompts, demonstrating the diversity and balance of our prompts.

## 4 TRAINING AND EVALUATION METHODS

### 4.1 SUPERVISED FINETUNING THE MLLM

As we mentioned above, we use MLLMs as the evaluation models and let it to answer a set of carefully-designed instructions, thereby achieving quantitative measurement of fine-grained text-to-image generation skills. Due to data bias, zero-shot MLLMs perform poorly when it comes to evaluation on generated images, particularly in term of image faithfulness. To solve this problem, we apply SFT training on the detailed human annotation to align the MLLM with human preference prior. Formally, the SFT training sample can be denoted as a triplet: question (or the instruction), multimodal input and answer. During SFT training, the optimization objective is the autoregressive loss function utilized to train LLMs, but calculated only on the answer, the loss function can be formulated as follows:

$$L(\theta) = \sum_{i=1}^{N} \log p(A_i|Q, M, A_{<i}; \theta), \tag{1}$$

where $N$ is the length of the ground truth answer, $Q$ is a fine-grained question of the generated image and its available answer, $M$ is the image and textual description, while $A$ is the human annotated answer selected from the given options. Notably, we expand each option to make it more detailed and descriptive, thereby benefiting SFT performance by allowing the MLLM to better understand the meaning of each option.

### 4.2 EVALUATION AND METRICS

To evaluate synthesized images with consideration of its synthetic nature, EVALALIGN is designed to evaluate image faithfulness and text-image alignment in a fine-grained way. Notably, image faithfulness and text-image alignment are two common errors occurred in synthesized images, whereas real images inherently exhibit high levels of both image faithfulness and text-image alignment.

**Image Faithfulness** measures whether synthesized images are faithful to real-world commonsense. With higher image faithfulness, the visual elements of generated images more closely resemble their real-world counterparts. Unfortunately, text-to-image models often generate images with low faithfulness, such as distorted body structures and human hands. This is also a critical reason why we set image faithfulness as one of the benchmarking aspects in EVALALIGN. Additionally, evaluating image faithfulness requires considering the input prompts, as prompts may describe unreal or impossible scenarios that inherently affect the faithfulness of the generated images. For example, when prompts like "a dog walking like a human" or "a man on Mars without a spacesuit" are provided, the generated images may naturally deviate from real-world image faithfulness. Under

such circumstances, the synthesized images cannot be regarded as low in image faithfulness since the generative models are merely following prompts that contain super-reality scenarios.

**Text-Image Alignment** evaluates whether generated images are aligned with their conditioned prompts. In the inference settings of text-to-image models, the image generation process is conditioned on textual prompts, requiring alignment between the text prompts and the synthesized images. However, through our observations, text-to-image models cannot consistently follow input prompts, often yielding images with visual elements misaligned with the input prompts. For example, models may generate images featuring an orange cat when conditioned on the text prompt "a blue cat."

During inference, the multimodal large language model (MLLM) is required to generate an appropriate response given a specific question $Q$ and multimodal input $M$ in an autoregressive manner:

$$R_i = f(Q, M, R_{<i}; \theta), \tag{2}$$

where $R_i$ is the $i$-th generated token, $R_{<i}$ represents the sequence of tokens generated before step $i$, and $\theta$ denotes the parameters of the fine-tuned MLLM. This autoregressive generation process is considered complete once the model generates an end-of-sequence (EOS) token or the generated response exceeds a preset maximum generation length. After generation, we employ rule-based filtering and regular expressions to extract the option chosen by the MLLM. Each option is assigned a unique predefined score to quantitatively measure a fine-grained skill specified by the question $Q$:

$$\text{Score}(Q) = g(R) = g(f(Q, M; \theta)), \tag{3}$$

where $g(\cdot)$ represents the procedure of option extraction and score mapping.

We devise two holistic and detailed question sets, $S_f$ and $S_a$, that encompass every aspect of image faithfulness and text-image alignment, respectively. Consequently, our metric, **EvalAlign**, can be defined by averaging the scores of the questions in the two sets:

$$\text{EvalAlign}_f = \frac{1}{|S_f|} \sum_{Q_i \in S_f} \text{Score}(Q_i), \tag{4}$$

$$\text{EvalAlign}_a = \frac{1}{|S_a|} \sum_{Q_j \in S_a} \text{Score}(Q_j), \tag{5}$$

where $\text{EvalAlign}_f$ and $\text{EvalAlign}_a$ indicate the image faithfulness score and the text-image alignment score evaluated by our method, respectively.

### 4.3 IMPLEMENTATION DETAILS

For details about the SFT training, we apply LoRA (Hu et al., 2021) finetuning on LLaVA-NeXT (Liu et al., 2024a) models to align them with the EVALALIGN dataset. Additionally, we merely adapt LoRA finetuning on the Q and K weights of the attention module, as extending the finetuning to the ViT (Dosovitskiy, 2020) and projection modules will lead to overfitting. The entire training process is conducted on 32 NVIDIA A100 GPUs for 10 hours, with a learning rate of $5 \times 10^{-5}$. As for the ablation study, we evaluate the finetuned LLaVA-NeXT 13B model on the validation dataset. In the final experiment, we apply SFT to the LLaVA-NeXT 34B model on the testing dataset to testify its generalization ability.

## 5 EXPERIMENTAL RESULTS

### 5.1 MAIN RESULTS

**Evaluation on image faithfulness.** We evaluate image faithfulness on the testing dataset to ensure that the finetuned MLLM aligns with human judgment and generalizes to unseen data. As detailed in Table 2, the finetuned MLLM successfully aligns with human preferences on image faithfulness, indicating its ability of image faithfulness evaluation is close to human. Specifically, the rankings of the top and bottom 10 models by both EVALALIGN and human evaluation scores are remarkably consistent. Besides, most of the images in the testing dataset, especially those from the 16 exclusive generative models, are not present during the SFT training, showcasing the robust generalization capability of our models.

Table 2: **Results on image faithfulness.** We evaluate the image faithfulness of images generated by 24 text-to-image models to compare five evaluation metrics against human scoring results. The experiments show that our metric's scores align more closely with human evaluations than those of other metrics.

| Model | Human | EVALALIGN | HPS v2 | CLIP-score | ImageReward | PickScore |
|---|---|---|---|---|---|---|
| PixArt XL2 1024 MS (Chen et al., 2023b) | 2.2848 [1] | 1.6415 [1] | 31.6226 [1] | 0.8580 [1] | 0.9696 [1] | 22.1335 [1] |
| Dreamlike Photoreal v2.0 (dreamlike.art, b) | 2.0070 [2] | 1.4522 [4] | 29.2322 [6] | 0.8286 [12] | 0.1886 [13] | 21.2271 [8] |
| SDXL Refiner v1.0 (Stability AI, b) | 1.9229 [3] | 1.6072 [2] | 29.8197 [3] | 0.8566 [2] | 0.7245 [2] | 22.0492 [2] |
| SDXL v1.0 (Podell et al., 2023) | 1.8136 [4] | 1.4675 [3] | 29.0620 [7] | 0.8467 [4] | 0.7043 [3] | 21.8106 [3] |
| Wuerstchen (Pernias et al., 2023) | 1.7837 [5] | 1.4279 [5] | 30.6622 [2] | 0.8199 [14] | 0.3212 [11] | 21.3720 [6] |
| LCM SDXL (Luo et al., 2023) | 1.6910 [6] | 1.3391 [7] | 29.3588 [5] | 0.8335 [10] | 0.5304 [6] | 21.6532 [4] |
| Openjourney (PromptHero, a) | 1.6667 [7] | 1.1750 [10] | 26.3475 [13] | 0.8196 [15] | 0.1478 [16] | 20.8637 [10] |
| Safe SD MAX (Patrick et al., 2022) | 1.6491 [8] | 1.2175 [8] | 25.7396 [17] | 0.7555 [24] | -0.0507 [22] | 20.4594 [21] |
| LCM LORA SDXL (Luo et al., 2023) | 1.6387 [9] | 1.3833 [6] | 27.3299 [10] | 0.8364 [8] | 0.4959 [7] | 21.4824 [5] |
| Safe SD STRONG (Patrick et al., 2022) | 1.6308 [10] | 1.1466 [11] | 25.5764 [18] | 0.8165 [18] | -0.1022 [23] | 20.6211 [18] |
| Safe SD MEDIUM (Patrick et al., 2022) | 1.6275 [11] | 1.1298 [15] | 26.2798 [14] | 0.8101 [20] | 0.2042 [12] | 20.7880 [12] |
| Safe SD WEAK (Patrick et al., 2022) | 1.6078 [12] | 1.1188 [17] | 26.1180 [15] | 0.7809 [23] | -0.1264 [24] | 20.3873 [24] |
| SD v2.1 (Rombach et al., 2022) | 1.5524 [13] | 1.1094 [18] | 26.5823 [12] | 0.8377 [7] | 0.4116 [9] | 21.0502 [9] |
| SD v2.0 (Rombach et al., 2022) | 1.5277 [14] | 1.1300 [14] | 25.3481 [21] | 0.8170 [17] | 0.0872 [18] | 20.7529 [13] |
| Openjourney v2 (PromptHero, b) | 1.5000 [15] | 0.9956 [20] | 24.6984 [23] | 0.7958 [22] | -0.0415 [21] | 20.4088 [22] |
| Redshift diffusion (Redshift-Diffusion) | 1.4733 [16] | 1.1382 [12] | 25.1572 [22] | 0.8101 [21] | 0.0218 [20] | 20.6155 [19] |
| Dreamlike Diffusion v1.0 (dreamlike.art, a) | 1.4652 [17] | 1.2052 [9] | 29.6506 [4] | 0.8543 [3] | 0.6508 [4] | 21.2664 [7] |
| SD v1.5 (Rombach et al., 2022) | 1.4417 [18] | 1.1362 [13] | 25.4972 [19] | 0.8214 [13] | 0.1686 [14] | 20.7143 [16] |
| IF-I-XL v1.0 (Alex Shonenkov & et al., 2023) | 1.3808 [19] | 0.9221 [22] | 27.4512 [9] | 0.8449 [5] | 0.6087 [5] | 20.7474 [14] |
| SD v1.4 (Rombach et al., 2022) | 1.3592 [20] | 0.9511 [21] | 25.3697 [20] | 0.8190 [16] | 0.1050 [17] | 20.6535 [17] |
| Vintedois Diffusion v0.1 (Vintedois-Diffusion v0.1) | 1.3562 [21] | 1.0797 [19] | 26.5901 [11] | 0.8341 [9] | 0.3562 [10] | 20.8358 [11] |
| IF-I-L v1.0 (Alex Shonenkov & et al., 2023) | 1.2635 [22] | 0.8814 [23] | 27.4836 [8] | 0.8384 [6] | 0.4463 [8] | 20.7170 [15] |
| MultiFusion (Marco et al., 2023) | 1.2372 [23] | 1.1298 [16] | 23.8133 [24] | 0.8151 [19] | 0.0695 [19] | 20.4780 [20] |
| IF-I-M v1.0 (Alex Shonenkov & et al., 2023) | 1.0135 [24] | 0.7928 [24] | 25.9522 [16] | 0.8329 [11] | 0.1637 [15] | 20.4035 [23] |

**Evaluation on text-image alignment.** The evaluation of text-image alignment on the testing dataset is similar to that of image faithfulness. Table 2 reveals that the rankings of the 24 evaluated models by EVALALIGN are generally consistent with human annotators. We believe that the consistency on image faithfulness and text-image alignment evaluations mainly stems from our annotated high-quality SFT dataset. It also proves that, with the annotated dataset and the extraordinary image-text joint understanding ability owned by MLLMs, we can easily finetune a MLLM to conduct the evaluation with low cost but close-to-human performance.

## 5.2 ABLATIONS AND ANALYSES OF EVALALIGN

**Results on different prompt categories.** Since MLLMs are not specifically trained to perform evaluations, they are naturally ill-suited for this task, hindering their task performances. Therefore, we need to annotate SFT data for this task and finetune the MLLMs accordingly. To verify the necessity, We conduct experiments comparing the LLava-Next 13B model with and without SFT. As shown in Table 4 and Table 5, the results demonstrate that SFT training considerably improves performance across all prompt categories in both image faithfulness and text-to-image alignment, closely aligning the MLLM's predictions with human evaluations. Note that Table 4 illustrates that the baseline method without SFT performs poorly in image faithfulness and text-image alignment evaluations, particularly in the former.

**Effect of training dataset size for vision-language model training.** In order to explore the effects of data size and determine the sufficient amount of training data, we train the model on image faithfulness evaluation task with images and their annotations sourced from 200, 500 and 800 prompts. As illustrated in Table 6, the evaluation performance continuously enhances as more training data is used. Notably, training with just 500 prompts nearly maximizes accuracy, with further increases to 800 data yielding only marginal improvements. This result suggests that our method requires only a small amount of annotated data to achieve good performance, highlighting its

Table 3: **Results on text-to-image alignment.** We evaluated the text-image alignment of images generated by 24 text-to-image models to compare how five evaluation metrics align with human scoring results. The experiments reveal that, in terms of text-image alignment metrics, our metric scores are highly consistent with human scores, demonstrating a much closer alignment than other evaluation metrics.

| Model | Human | EVALALIGN | HPS v2 | CLIP-score | ImageReward | PickScore |
|---|---|---|---|---|---|---|
| IF-I-XL v1.0 (Alex Shonenkov & et al., 2023) | 5.4500[1] | 5.5300[1] | 32.5477[10] | 0.8579[2] | 0.4391[3] | 21.1998[10] |
| IF-I-L v1.0 (Alex Shonenkov & et al., 2023) | 5.2300[2] | 5.4500[2] | 32.7140[9] | 0.8538[4] | 0.3820[6] | 21.1284[12] |
| SDXL Refiner v1.0 (Stability AI, b) | 5.2100[3] | 5.4000[3] | 35.6465[3] | 0.8528[5] | 0.4738[2] | 22.3532[2] |
| LCM SDXL (Luo et al., 2023) | 5.1800[4] | 5.3300[5] | 33.8011[6] | 0.8512[6] | 0.3833[5] | 21.9620[4] |
| PixArt XL2 1024 MS (Chen et al., 2023b) | 5.1100[5] | 5.3100[6] | 37.0493[1] | 0.8634[1] | 0.6542[1] | 22.3926[1] |
| IF-I-M v1.0 (Alex Shonenkov & et al., 2023) | 5.0800[6] | 5.2200[8] | 31.0951[14] | 0.8434[8] | 0.0499[10] | 20.8270[20] |
| LCM LORA SDXL (Luo et al., 2023) | 5.0600[7] | 5.2700[7] | 32.7752[8] | 0.8349[10] | 0.1618[9] | 21.7627[6] |
| SDXL v1.0 (Podell et al., 2023) | 5.0300[8] | 5.3500[4] | 35.1593[4] | 0.8540[3] | 0.4322[4] | 22.1291[3] |
| Wuerstchen (Pernias et al., 2023) | 4.8700[9] | 5.1700[9] | 36.4632[2] | 0.8381[9] | 0.2513[7] | 21.7779[5] |
| Openjourney (PromptHero, a) | 4.8300[10] | 4.9200[15] | 31.1495[12] | 0.8173[16] | -0.0867[14] | 21.1163[13] |
| SD v2.1 (Rombach et al., 2022) | 4.8000[11] | 5.0700[11] | 31.1017[13] | 0.8278[14] | -0.0453[12] | 21.2093[9] |
| MultiFusion (Marco et al., 2023) | 4.6800[12] | 4.8000[18] | 28.7957[24] | 0.8264[15] | -0.1337[15] | 20.9625[17] |
| Dreamlike Diffusion v1.0 (dreamlike.art, a) | 4.6600[13] | 5.1500[10] | 34.8196[5] | 0.8493[7] | 0.2295[8] | 21.5550[7] |
| SD v2.0 (Rombach et al., 2022) | 4.6400[14] | 5.0100[12] | 30.6153[17] | 0.8298[13] | -0.1424[16] | 21.1905[11] |
| Vintedois Diffusion v0.1 (Vintedois-Diffusion v0.1) | 4.6200[15] | 4.9500[14] | 31.9503[11] | 0.8319[12] | -0.0222[11] | 21.1141[14] |
| Safe SD STRONG (Patrick et al., 2022) | 4.6000[16] | 4.8300[17] | 30.6615[16] | 0.7751[23] | -0.5028[22] | 20.7491[21] |
| Dreamlike Photoreal v2.0 (dreamlike.art, b) | 4.5600[17] | 4.9800[13] | 33.7712[7] | 0.8344[11] | -0.0859[13] | 21.4832[8] |
| Safe SD WEAK (Patrick et al., 2022) | 4.5300[18] | 4.7100[20] | 30.5644[18] | 0.8140[18] | -0.2728[18] | 20.9899[16] |
| SD v1.4 (Rombach et al., 2022) | 4.5200[19] | 4.7600[19] | 29.9149[20] | 0.8048[20] | -0.3438[19] | 20.8462[19] |
| SD v1.5 (Rombach et al., 2022) | 4.4500[20] | 4.9000[16] | 30.1673[19] | 0.8142[17] | -0.2213[17] | 20.8640[18] |
| Safe SD MEDIUM (Patrick et al., 2022) | 4.4000[21] | 4.5600[24] | 30.7820[15] | 0.7974[21] | -0.3591[20] | 21.0257[15] |
| Redshift diffusion (Redshift-Diffusion) | 4.3500[22] | 4.6700[21] | 29.2865[22] | 0.8066[19] | -0.4172[21] | 20.6327[23] |
| Safe SD MAX (Patrick et al., 2022) | 4.3100[23] | 4.5900[23] | 29.8126[21] | 0.7601[24] | -0.6095[24] | 20.7046[22] |
| Openjourney v2 (PromptHero, b) | 4.1500[24] | 4.6500[22] | 29.2389[23] | 0.7851[22] | -0.6051[23] | 20.5973[24] |

Table 4: **Results of different prompt categories for evaluating image faithfulness.** Baseline is the vanilla LLaVA-NeXT model without find-tuning with human-aligned data.

| Method | Body | Hand | Face | Object | Common |
|---|---|---|---|---|---|
| Human | 1.6701 | 1.0278 | 1.4107 | 2.2968 | 1.0637 |
| Baseline | 3.9950 | 3.9932 | 3.9867 | 2.6734 | 3.3476 |
| EVALALIGN | 1.7305 | 0.9490 | 1.4393 | 2.3565 | 1.0903 |

Table 5: **Results of different prompt categories for evaluating text-to-image alignment.** Baseline is the vanilla LLaVA-NeXT model without find-tuning with human-aligned data.

| Method | Object | Count | Color | Style | Spatial | Action |
|---|---|---|---|---|---|---|
| Human | 1.6947 | 1.2032 | 1.8551 | 1.9796 | 1.5608 | 1.8015 |
| Baseline | 1.5602 | 1.0742 | 1.9275 | 1.1837 | 1.4118 | 1.1838 |
| EVALALIGN | 1.6807 | 1.2516 | 1.8696 | 1.9592 | 1.5882 | 1.8382 |

cost-effectiveness. Generally, since more data leads to better performance, we use all of the available data to finetune our models and release this data to the research community to bootstrap further study.

**Effect of model size.** Since transformers are known for their scalability (Radford et al., 2018; Dehghani et al., 2023), we investigate the effect of the model size on the performance of image faithfulness evaluation. As illustrated in Table 7, the benefits of scaling up the utilized MLLMs are remarkably significant, where increasing the model size from 7B to 34B results in substantial improvements in evaluation performance. For this consequence, for the final version of the EVALALIGN evaluation model, we choose LLaVA-NeXT 34B, the largest model in LLaVA-NExT series, and finetune it on our meticulously curated SFT data. Since some users of EVALALIGN cannot afford MLLM inference with 34B parameters, we will make the 13B and 34B models publicly available.

5.3 COMPARISON WITH EXISTING EVALUATION METHODS

**SFT with human-aligned data outperforms vanilla MLLMs.** To validate the effectiveness of the MLLM after SFT, we use vanilla LLaVA-NeXT 13B as the baseline model for comparison. As shown

Table 6: **Ablation study on the size of training data.** Results are reported on image faithfulness under different training data scale. We observe that a small number of annotated training data is sufficient for optimal results.

| Method | Data Size | SDXL | Pixart | Wuerstchen | SDXL-Turbo | IF | SD v1.5 | SD v2.1 | LCM |
|---|---|---|---|---|---|---|---|---|---|
| Human | – | 2.1044 | 1.8606 | 1.7839 | 1.3854 | 1.3822 | 1.3818 | 1.1766 | 1.0066 |
| EVALALIGN | 200 | 1.7443 | 1.8898 | 1.9278 | 1.1261 | 1.2977 | 1.5254 | 1.4309 | 1.1204 |
| | 500 | 1.8890 | 1.9161 | 1.8586 | 1.2141 | 1.3109 | 1.3926 | 1.3815 | 0.9485 |
| | 800 | 2.0443 | 1.9199 | 1.8012 | 1.3353 | 1.296 | 1.4702 | 1.3221 | 1.0305 |

Table 7: **Ablation study on the size vision-language model.** Results are reported on image faithfulness under different model scales of LLaVA-NeXT. We observe that model size is critical for reliable evaluation.

| Method | Model Size | SDXL | Pixart | Wuerstchen | SDXL-Turbo | IF | SD v1.5 | SD v2.1 | LCM |
|---|---|---|---|---|---|---|---|---|---|
| Human | – | 2.1044 | 1.8606 | 1.7839 | 1.3854 | 1.3822 | 1.3818 | 1.1766 | 1.0066 |
| EVALALIGN | 7B | 1.9959 | 1.8615 | 1.8228 | 1.1708 | 1.2704 | 1.4031 | 1.3063 | 1.0145 |
| | 13B | 2.0443 | 1.9199 | 1.8012 | 1.3353 | 1.2960 | 1.4702 | 1.3221 | 1.0305 |
| | 34B | 2.1131 | 1.8621 | 1.8083 | 1.3906 | 1.3076 | 1.3921 | 1.2037 | 1.0143 |

in Table 4 and Table 5, the results of vanilla model suggest some correlations with human-annotated data. However, the alignment of the vanilla MLLM is relatively low due to the absence of images generated by model (such as distorted bodies and hands images) and issues related to evaluation in the MLLM's pre-training dataset. After applying SFT on the LLaVA-Next 13B model using human annotated data, the model's predictions on various fine-grained evaluation metrics are almost align to the human-annotated data and significantly surpass the evaluation results of all MLLM models that are not finetuned. This experimental results confirms that our SFT training enables the MLLM to be successfully applied to the task of evaluating text-to-image models.

**Comparison with other methods.** To verify the human preference alignment of our model, especially when compared with other baseline methods, we calculate Kendall rank (KENDALL, 1938) and Pearson (Freedman et al., 2007) correlation coefficient on images generated by 24 text-to-image models and summarize the results in Table 8.

As can be concluded, compared with baseline methods, EVALALIGN achieves significant higher alignment with fine-grained human preference on image faithfulness and image-text consistency, showcasing robust generalization ability. Although HPS v2 roughly aligns with human preference in some extent, the relative small model capacity and coarse ranking training limits its generalization to the fine-grained annotated data. Besides, since CLIP-s only cares the CLIP similarity of the generated image and its corresponding prompt, it behaves poorly in image faithfulness evaluation. The per-question alignment and the leaderboard of EVALALIGN will be introduced in the supplementary materials.

Table 8: **Comparison with existing methods.**

| Method | Faithfulness | | Alignment | |
|---|---|---|---|---|
| | Kendall↑ | Pearson↑ | Kendall↑ | Pearson↑ |
| CLIP-score | 0.1304 | 0.1765 | 0.6956 | 0.8800 |
| HPSv2 | 0.4203 | 0.5626 | 0.5217 | 0.7113 |
| EVALALIGN | 0.7464 | 0.8730 | 0.8043 | 0.9356 |

## 6 CONCLUSION AND DISCUSSION

In this work, we design an economic evaluation method that offers high accuracy, strong generalization capabilities, and provides fine-grained, interpretable metrics. We develop a comprehensive data annotation and cleaning process tailored for evaluation tasks, and establish the EVALALIGN benchmark for training and evaluating models on supervised fine-tuning tasks for MLLMs. Experimental results across 24 text-to-image models demonstrate that our evaluation metrics surpass the accuracy of all the state-of-art evaluation method. Additionally, we conduct a detailed empirical study on how MLLMs can be applied to model evaluation tasks. There are still many opportunities for further advancements and expansions based on our EVALALIGN. We hope that our work can inspire and facilitate future research in this field.

## 7 REPRODUCIBILITY STATEMENT

The full version of the source code, dataset, as well as the final version of the finetuned MLLMs (one finetuned on LLaVA-NeXT 13B and the other one finetuned on LLaVA-NeXT 34B) will be released to the public. The data construction procuedure, including data collection and curation, data cleaning and annotation, is thoroughly described in Section 3. For details related to the human annotation and the measures that used to ensure its quality, we comprehensively introduce them in Appendix B. As for every experiment introduced in this paper, we provide a general introduction in Section 5 and exhibit implementation details related to reproduce our experiments. Specifically, the latter includes the hyper-parameters of each evaluated models, the employed instruction, as well as more supplementary experiments, which are described in Appendix C, Appendix D and Appendix E.

## 8 ETHICS STATEMENT

We are committed to conducting this research with the highest ethical standards. Our goal is to contribute positively to the fields of evaluation benchmarks on artificial intelligence generated content, emphasizing transparency and reproducibility in our design. Similar with other MLLMs, EVALALIGN may potentially generate responses contain offensive, inappropriate, or harmful content. Since the base MLLMs of EVALALIGN are pretrained on large datasets scraped from the web that might contain private information and harmful content, they may inadvertently generate or expose sensitive information, raising ethical and privacy concerns. MLLMs are also susceptible to adversarial attacks, where inputs are intentionally crafted to deceive the model. This vulnerability can be exploited to manipulate model outputs, posing security and ethic risks. To alleviate these safety limitation and our fulfill our social responsibility as artificial intelligence researchers, we create dedicated evaluation sets for bias detection and mitigation, and conducted adversarial testing through hours of redteaming. Besides, EVALALIGN is designed for fine-grained, human-aligned automatic text-to-image evaluations, which can serve as a stepping stones toward revealing the inner generation nature of text-to-image generative models, thereby lowering the ethical hazard of these models. We believe that with appropriate use, it could provide users with interesting experiences for detailed synthesized image evaluation, and inspires more appealing research works about text-to-image generation.

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
