# EvalAlign: Supervised Fine-Tuning Multimodal LLMs with Human-Aligned Data for Evaluating Text-to-Image Models (Supplementary Materials)

## Contents

## A  Limitations

**Multimodal LLMs.** Since EvalAlign evaluation models are fine-tuned MLLMs, they also suffer from multimodal hallucination, where models may generate content that seems plausible but actually incorrect or fabricated, and cannot be inferred from the input images and texts. Moreover, due to the possible harmful content in the pretraining data of the utilized base MLLMs, the model may inherit these biases and generate inappropriate response. Although we carefully curate the SFT training data of the EvalAlign evaluation models, the problems of hallucination and biased pre-training is alleviated but not fully addressed. Other than the these issues, EvalAlign evaluation models also suffer from opacity and interpretability, context limitation, as well as sensitivity to input formatting, like most multimodal LLMs.

**Human Annotations.** Human annotation is naturally subjective and influenced by individual perspectives, biases, and preferences. During the annotation, annotators can make mistakes, leading to incorrect or noisy labels. Regarding these challenges, we conduct 9 rounds of trial annotation and 2 rounds of random sampling quality inspection to ensure the inter-annotator consistency and overall annotation quality. We also design easy-to-understand annotation guidelines, instructions and platform to lower the annotation difficulty and benefit the annotation accuracy. Despite all these efforts, conducting human annotation with different annotators, user interface and annotation guidelines may lead to different result, making our annotation somewhat limited. Furthermore, human annotation can be time-consuming and resource-intensive, limiting the scale at which we can afford.

No.4512 Prompt: **A woman wearing a red dress is gently cradling a small, white puppy in her arms.**

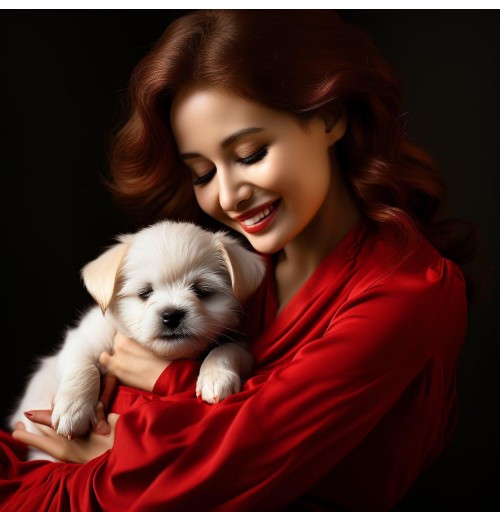

Q2:Are there any issues with the [human/animals] hands in the image?

1.The hand in the picture has a very grievous problem that is unbearable;

2.The hand in the picture has some serious problems and is not acceptable;

3.The hand in the picture has a slight problem that does not affect the senses;

4.The hand in the picture is basically fine, with only a few flaws;

5.The hands in the picture are completely fine and close to reality.

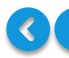 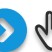 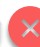

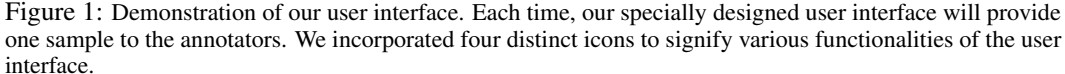

Figure 1: Demonstration of our user interface. Each time, our specially designed user interface will provide one sample to the annotators. We incorporated four distinct icons to signify various functionalities of the user interface.

## B  ANNOTATION DETAILS

Before performing the final human annotation, we made a series of efforts to guarantee its quantity, quality and efficiency. To begin with, we select appropriate candidates to perform the annotation and hold a training meeting for them. Then, we design a user-friendly user interface and a comprehensive annotation procedure. We write a detailed annotation guidelines to explain every aspect and precaution of the annotation. As mentioned above, we conduct 9 rounds of trial annotation on another 50 synthesized images and 2 turns of random sampling quality inspection to further ensure inter-annotator consistency and annotation accuracy.

**Annotator selection.** The accuracy and reliability of the annotated data depend heavily on the capabilities of the human annotators involved in the annotation process. As a consequence, at the beginning of the annotation, We first conduct annotator selection to build an appropriate and unbiased annotation team, and train this annotation team with our meticulously prepared annotation guidelines. For annotator selection, we let the candidates to accomplish a test concentrating on 10 factors, domain expertise, resistance to visually disturbing content, attention to detail, communication skills, reliability, cultural and linguistic competence, technical skills, ethical considerations, aesthetic cognition, and motivation. Notably, since the evaluated models may generate images with uncomfortable and inappropriate visual content, the candidates are notified with this inconvenience before the test. Only those agreed with this inconvenience are eligible to participate in the test, and they are welcome to withdraw at any time if they choose to do so. Based on the test results and candidate backgrounds, We try our best to ensure that the selected annotators are well-balanced in background and have a generally competitive abilities of the 10 mentioned factors. To summarize, our annotation team includes 10 annotators carefully selected from 29 candidates, 5 males and 5 females, all have a bachelor's degree. We interview the annotators and ensure they are adequate for the annotation.

**Annotation training and guidelines.** After the selection, we conduct a training meeting over our comprehensive user guidelines to make the annotation team aware of our purpose and standard. During the training meeting, we explain the purpose, precaution, standard, workload and wage of the annotation. Besides, we have formally informed the annotators that the human annotation is solely for research purposes, and the data they have annotated may potentially be released to the public in the future. We, and the annotators reached consensus on the standard, workload, wage and intended usage of the annotated data. The rules for recognising image faithfulness and text-image

are universal, and thus each individual's standards should not differ significantly. As a consequence, we annotate a few samples using our meticulously developed annotation platform for the annotators to ensure inter-annotator consistency. The overall snapshot of the developed annotation paltform is exhibited in fig. 1. With this training, we also equip the annotators with necessary knowledge for unbiased detailed human evaluation on image faithfulness and text-image alignment. Specifically, the employed annotation guidelines involve the instructions for using the annotation platform and detailed guidelines about the annotation procedure, and we demonstrate them in Table 1.

**Trial Annotation** Even with the above preparation, there is no quantitative evidence to verify the quality, the efficiency, and the inter-annotator consistency of the human annotation. Additionally, the standard for assessing image faithfulness and text-image are universal, which further emphasize the significant role of high inter-annotator consistency. Considering that, we conduct a multi-turn trial annotation on another 50 synthesized images. After each trial, we calculate the Cohen's kappa coefficient and conduct a meeting for our annotators to explain annotation standards, rules and guidelines, thereby ensuring high inter-annotator reliability. In total, we conduct nine turns of trial annotation, and in the last turn of the trial, the Cohen's kappa coefficient of our annotators reaches 0.681, indicating high inter-annotator reliability.

**Random Sampling Quality Inspection** Upon reaching the milestone percentages of 25%, 50%, 75%, and 100% in the annotation progress, we conducted a series of random sampling quality inspections on the present annotation results at each milestone, totally four turns of random sampling quality inspection. The random sampling quality inspection by four experts in text-to-image generation selected from our group on 1,000 randomly sampled annotated images. For the first two turn of quality inspection, there are totally 423 and 112 annotated samples that failed the inspection. The failed samples are re-annotated and re-inspected. As for the last two turns of quality inspection, they both revealed zero failed samples due to the thoughtful and rigorous annotation preparation.

## C  ADDITIONAL DETAILS OF THE EVALUATED MODELS

In this section, we introduce the details of the evaluated text-to-image generative models in this work.

- **Stable Diffusion {v1.4, v1.5, v2 base, v2.0, v2.1}.** Stable Diffusion (SD) is a series of 1B text-to-image generative models based on latent diffusion model (Rombach et al., 2022) and is trained on LAION-5B (Schuhmann et al., 2022). Specifically, the SD series includes SD v1.1, SD v1.2, SD v1.4, SD v1.5, SD v2 base, SD v2.0, and SD v2.1 respectively. Among them, we choose the most commonly-employed SD v1.4, SD v1.5, SD v2.0 and SD v2.1 for EVALALIGN evaluation.

  SD v1.1 was trained at a resolution of 256x256 on laion2B-en for 237k steps, followed by training at a resolution of 512x512 on laion-high-resolution ((170M examples from LAION-5B with resolution >= 1024x1024) for the subsequent 194k steps. While, SD v1.2 was initialized from v1.1 and further finetuned for 515k steps at resolution 512x512 on laion-aesthetics v2 5+ (a subset of laion2B-en, filtered to images with an original size >= 512x512, estimated aesthetics score > 5.0, and an estimated watermark probability < 0.5). **SD v1.4** is initialized from v1.2 and subsequently finetuned for 225k steps at resolution 512x512 on laion-aesthetics v2 5+. This version incorporates a 10% dropping of the text-conditioning to improve classifier-free guidance sampling. Similar to SD v1.4, **SD v1.5** is resumed from SD v1.2 and trained 595k steps at resolution 512x512 on laion-aesthetics v2 5+, with 10% dropping of the text-conditioning.

  SD v2 base is trained from scratch for 550k steps at resolution 256x256 on a subset of LAION-5B filtered for explicit pornographic material, using the LAION-NSFW classifier with punsafe = 0.1 and an aesthetic score >= 4.5. Then it is further trained for 850k steps at resolution 512x512 on the same dataset on images with resolution >= 512x512. **SD v2.0** is resumed from stable-diffusion v2 base and trained for 150k steps using a v-objective on the same dataset. After that, it is further finetuned for another 140k steps on 768x768 images. **SD v2.1** is finetuned from SD v2.0 with an additional 55k steps on the same dataset (with punsafe=0.1), and then finetuned for another 155k extra steps with punsafe=0.98.

- **Stable Diffusion XL {v1.0, Refiner v1.0}.** Stable Diffusion XL (SDXL) is a powerful text-to-image generation model that iterates on the previous Stable Diffusion models in three key ways: (1) its UNet is 3x larger and SDXL combines a second text encoder (OpenCLIP ViT-bigG/14)

with the original text encoder to significantly increase the number of parameters; (2) it introduces size and crop-conditioning to preserve training data from being discarded and gain more control over how a generated image should be cropped; (3) it introduces a two-stage model process; the base model (can also be run as a standalone model) generates an image as an input to the refiner model which adds additional high-quality details.

- **Pixart-Alpha.** Pixart-Alpha is a model that can be used to generate and modify images based on text prompts. It is a Transformer Latent Diffusion Model that uses one fixed, pretrained text encoders (T5)) and one latent feature encoder (VAE).

- **Latent Consistency Model Stable Diffusion XL** Latent Consistency Model Stable Diffusion XL (LCM SDXL) Luo et al. (2023) enables SDXL for swift inference with minimal steps. Viewing the guided reverse diffusion process as solving an augmented probability flow ODE (PF-ODE), LCMs are designed to directly predict the solution of such ODE in latent space, mitigating the need for numerous iterations and allowing rapid, high-fidelity sampling.

- **Dreamlike Diffusion 1.0.** Dreamlike Diffusion 1.0 (dreamlike.art, a) is a SD v1.5 model finetuned on high-quality art images by dreamlike.art.

- **Dreamlike Photoreal 2.0.** Dreamlike Photoreal 2.0 (dreamlike.art, b) is a photorealistic text-to-image latent diffusion model resumed from SD v1.5 by dreamlike art. This model was finetuned on 768x768 images, it works pretty good with resolution 768x768, 640x896, 896x640 and higher resolution such as 768x1024.

- **Openjourney v1, v2.** Openjourney (PromptHero, a) is an open-source text-to-image generation model resumed from SD v1.5 and finetuned on Midjourney images by PromptHero. Openjourney v2 (PromptHero, b) was further finetuned using another 124000 images for 12400 steps, about 4 epochs and 32 training hours.

- **Redshift Diffusion.** Redshift Diffusion (Redshift-Diffusion) is a Stable Diffusion model finetuned on high-resolution 3D artworks.

- **Vintedois Diffusion.** Vintedois Diffusion (Vintedois-Diffusion v0.1) is a Stable Diffusion v1.5 model finetuned on a large number of high-quality images with simple prompts to generate beautiful images without a lot of prompt engineering.

- **Safe Stable Diffusion {Weak, Medium, Strong, Max}.** Safe Stable Diffusion (Patrick et al., 2022) is an enhanced version of the SD v1.5 model by mitigating inappropriate degeneration caused by pretraining on unfiltered web-crawled datasets. For instance SD may unexpectedly generate nudity, violence, images depicting self-harm, and otherwise offensive content. Safe Stable Diffusion is an extension of Stable Diffusion that drastically reduces this type of content. Specifically, it has an additional safety guidance mechanism that aims to suppress and remove inappropriate content (hate, harassment, violence, self-harm, sexual content, shocking images, and illegal activity) during image generation. The strength levels for inappropriate content removal are categorized as: {Weak, Medium, Strong, Max}.

- **MultiFusion.** MultiFusion (Marco et al., 2023) is a multimodal, multilingual diffusion model that extends the capabilities of SD v1.4 by integrating various modules to transfer capabilities to the downstream model. This combination results in novel decoder embeddings, which enable prompting of the image generation model with interleaved multimodal, multilingual inputs, despite being trained solely on monomodal data in a single language.

- **DeepFloyd-IF { M, L, XL } v1.0.** DeepFloyd-IF (Alex Shonenkov & et al., 2023) is a novel state-of-the-art open-source text-to-image model with a high degree of photorealism and language understanding. It is a modular composed of a frozen text encoder and three cascaded pixel diffusion modules: a base model that generates 64x64 image based on text prompt and two super-resolution models, each designed to generate images of increasing resolution: 256x256 and 1024x1024, respectively. All stages of the model utilize a frozen text encoder based on the T5 transformer to extract text embeddings, which are then fed into a UNet architecture enhanced with cross-attention and attention pooling. Besides, it underscores the potential of larger UNet architectures in the first stage of cascaded diffusion models and depicts a promising future for text-to-image synthesis. The model is available in three different sizes: M, L, and XL. M has 0.4B parameters, L has 0.9B parameters, and XL has 4.3B parameters.

Table 1: User Guidelines of the Human Annotation. Considering that our annotators are native Chinese speakers while our readers may not be, each user is actually provided with a copy of Chinese version of the user guidelines. Meanwhile, we demonstrate its translated English version as follows.

| User Guidelines |
| --- |
| **Part I Introduction** |
| Welcome to the annotation platform. This platform is designed to simplify the annotation process and enhance annotation efficiency. Before the detailed introduction, we want to claim again that you may feel inconvenient as the evaluated models may generate images with uncomfortable and inappropriate visual content. Now, you are still welcomed if you want to withdraw your consent The annotation process is conducted on a sample-by-sample basis, with a question-by-question approach. Thus, you are supposed to answer all the questions raised for the present sample to accomplish its annotation. Once all the delegated samples are accomplished, your job is finished and we are thankful for your contribution to the project. |
| **Part II Guidelines of the User Interface** |
| 1.User Login: To access the annotation platform, you are required to login as a user. Please navigate to the login page, enter the username and password provided by us, and click the "Login" button. |
| 2.Dashboard: Once you complete the login, you will be jumped into the dashborad page. The dashboard will list the overview of the samples assigned to you to annotate. Besides, we list the status of each sample for you to freely check your annotation progress (e.g., pending, completed). |
| 3.Annotation Interface: Click on the "Start" button or an assigned image through the dashboard interface, you will jump into the annotation interface. annotation interface is made up of three components: 1) Image Display: View the image to be annotated and its conditioned prompts; 2) Question Panel: List of single-choice questions related to the image; 3) Navigation Buttons: "Next" and "Previous" buttons to navigate through questions and images. |
| 4.Answering Questions: Each time, the annotation interface will provide you a sample for annotation, please view the image and read the associated question, select the appropriate answer from the available options, and repeat the processs for all questions related to the question. |
| 5.Saving and Submitting Annotations: To save progress and submit completed annotations, you can click the "Save" button to save your progress. If you finish the assigned sample and ensure the accuracy and confidence of its, you can click the "Submit" button to submit this annotation. |
| 6.Review and Edit Annotations: If you want to review and edit your submission, you can navigate the completed tasks section, and select the image to review. You will jump into its annotation interface with the previously submitted annotations and are allowed to do any modification. |
| 7.Report and contact: If you find any problem about the assigned sample, such as witnessing NSFW or biased content, assigned visually abnormal sample, feel free to click the "Report" button and fill a form to report this sample. If you have any question about the standard of the annotation or have suggestions for improvement, please do not hesitate to contact us through phone, we will be glad to help you. |
| **Part III General Guidelines of the Human Annotation** |
| 1.In general, you are supposed to answer all the questions raised for the present sample to accomplish its annotation. This annotation only involves single-choice question. |
| 2.Before answering the question, please ensure that the question is applicable to this prompt. If it is not applicable, please select option 0 directly—this is the predefined option for this particular scenario. |
| 3.If you are answering a question about the image faithfulness, you may find the question is applicable to multiple objects within the image. you need to answer the question regarding to every applicable object and its role in the image. A straightforward way for this is to solely score every applicable object and choose the option closest to the calculated weighted average score. |
| 4.If you are answering the object faithfulness question on the image faithfulness annotation, you need to drop and report for the encountered image with no clear main object. |
| 5.If you are answering the commonsense question on the image faithfulness annotation, you need to drop and report for the encountered surreal and sci-fi image. |
| 6.You are required to first annotate 30 samples to form a stable and reasonable assessment standard. Then, accomplish the annotation in progress. |
| 7.This annotation is for evaluating image faithfulness and text-image alignment, as a consequence, the standard of the annotation is universal. |
| 8.If you feel confused at anything about the human annotation, feel free to contact us through phone, we will be glad to help you. |
| 9. Once you have submitted your annotation results, we are very thankful to inform you that you have finished your job. Thank you once again for your contribution to our project. |
| 10.If you have submitted your annotation but want to withdraw your submission and review the annotation results, you can contact us through phone, we will send it back to you. |

# D  INSTRUCTION TEMPLATES

Here, we present every instruction used for EVALALIGN evaluation on image faithfulness and text-image alignment. The templates contain some placeholders set for filling in the corresponding attributes of the input images during the evaluation. For example, a specific "<ObjectHere>" and "<NumberHere>" can be "people, laptop, scissors." and "plate: 1, turkey sandwich: 3, lettuce: 1.", respectively.

For EVALALIGN evaluation on image faithfulness, we devise 5 questions concentrate on the faithfulness of the generated body structure, generated face, generated hand, generated objects, as well as generation adherence to commonsense and logic. The instruction templates for these fine-grained criteria are as follows:

> **[Q1]:**Are there any issues with the [human/animals] body structure in the image, such as multiple arms, missing limbs or legs when not obscured, multiple heads, limb amputations, and etc?
> **[OPTIONS]**: 0.There are no human or animal body in the picture; 1.The body structure of the people or animals in the picture has a very grievous problem that is unbearable; 2.The body structure of the people or animals in the picture has some serious problems and is not acceptable; 3.The body structure of the people or animals in the picture has a slight problem that does not affect the senses; 4.The body structure of the people or animals in the picture is basically fine, with only a few flaws; 5.The body structure of the people or animals in the picture is completely fine and close to reality.

> **[Q2]:**Are there any issues with the [human/animals] hands in the image, such as having more or less than five fingers when not obscured, broken fingers, disproportionate finger sizes, abnormal nail size proportions, and etc?
> **[OPTIONS]**: 0.No human or animal hands are shown in the picture; 1.The hand in the picture has a very grievous problem that is unbearable; 2.The hand in the picture has some serious problems and is not acceptable; 3.The hand in the picture has a slight problem that does not affect the senses; 4.The hand in the picture is basically fine, with only a few flaws; 5.The hands in the picture are completely fine and close to reality.

> **[Q3]:**Are there any issues with [human/animals] face in the image, such as facial distortion, asymmetrical faces, abnormal facial features, unusual expressions in the eyes, and etc?
> **[OPTIONS]**: 0.There is no face of any person or animal in the picture; 1.The face of the person or animal in the picture has a very grievous problem that is unbearable; 2.The face of the person or animal in the picture has some serious problems and is not acceptable; 3.The face of the person or animal in the picture has a slight problem that does not affect the senses; 4.The face of the person or animal in the picture is basically fine, with only a few flaws; 5.The face of the person or animal in the picture is completely fine and close to reality.

> **[Q4]:**Are there any issues or tentative errors with objects in the image that do not correspond with the real world, such as distortion of items, and etc?
> **[OPTIONS]**: 0.There are objects in the image that completely do not match the real world, which is very serious and intolerable; 1.There are objects in the image that do not match the real world, which is quite serious and unacceptable; 2.There are slightly unrealistic objects in the image that do not affect the senses; 3.There are basically no objects in the image that do not match the real world, only some flaws; 4.All objects in the image match the real world, no problem.

**[Q5]**:Does the generated image contain elements that violate common sense or logical rules, such as animal/human with inconsistent anatomy, object-context mismatch, impossible physics, scale and proportion issues, temporal and spatial inconsistencies, hybrid objects, and etc?
**[OPTIONS]**: 0.The image contains elements that violate common sense or logical rules, which is very grievous and intolerable; 1.The presence of elements in the image that seriously violate common sense or logical rules is unacceptable; 2.The image contains elements that violate common sense or logical rules, which is slightly problematic and does not affect the senses; 3.There are basically no elements in the image that violate common sense or logical rules, only some flaws; 4.There are no elements in the image that violate common sense or logical rules, and they are close to reality.

The templates of EVALALIGN evaluation on text-image alignment are as follows. We select 6 common aspects of text-image alignment, object, number, color, style, spatial relationship and action. For images that do not involve the specified attribute, the corresponding question template is not filled in and subsequently input into EVALALIGN.

**[Q1]**:Does the given image contain all the objects (<ObjectHere>) presented in the corresponding prompts?
**[OPTIONS]**: 1.None objects are included; 2.Some objects are missing; 3.All objects are included.

**[Q2]**:Does the given image correctly reflect the numbers (<NumberHere>) of each object presented in the corresponding prompts?
**[OPTIONS]**: 1.All counting numbers are wrong; 2.Some of them are wrong; 3.All counting numbers are right.

**[Q3]**:Does the given image correctly reflect the colors of each object (<ColorHere>) presented in the corresponding prompts?
**[OPTIONS]**: 1.All colors are wrong; 2.Some of them are wrong; 3.All corresponding colors numbers are right.

**[Q4]**:Does the given image correctly reflect the style (<StyleHere>) described in the corresponding prompts?
**[OPTIONS]**: 1.All styles are wrong; 2.Some of them are wrong; 3.All styles are right.

**[Q5]**:Does the given image correctly reflect the spatial relationship (<SpatialHere>) of each object described in the corresponding prompts?
**[OPTIONS]**: 1.All spatial relationships are wrong; 2.Some of them are wrong; 3.All spatial relationships are right.

**[Q6]**:Does the given image correctly reflect the action of each object (<ActionHere>) described in the corresponding prompts?
**[OPTIONS]**: 1.All actions are wrong; 2.Some of them are wrong; 3.All actions are right.

# E    ADDITIONAL QUANTITATIVE ANALYSIS

## E.1    GENERALIZATION EXPERIMENTS

To verify the generalization capability of our evaluation model, We compared MLLM's SFT using different training datasets: one with images generated by all 8 text-to-image models and another with images generated by only 4 of these models, while the final evaluation was conducted on images generated by the other 4 models. As shown in Table 2 and Table 3, We observed that MLLMs trained on images from a subset of text-to-image models can effectively generalize to images generated by unseen text-to-image models.

Table 2: **Ablation study on the number of different text-to-image models used to generate the training data for evaluating image faithfulness.** We observe that EVALALIGN exhibits strong generalization capability.

| Method | T2I models | body | hand | face | object | common | MAE |
|--------|-----------|------|------|------|--------|--------|-----|
| Human | - | 1.4988 | 0.8638 | 1.1648 | 2.2096 | 0.8710 | 0 |
| EVALALIGN | 8 | 1.6058 | 0.7901 | 1.1974 | 2.2783 | 0.8871 | 0.0596 |
| | 4 | 1.6522 | 0.9588 | 1.2355 | 2.3032 | 0.9516 | 0.0987 |

Table 3: **Ablation study on the number of different text-to-image models used to generate the training data for evaluating text-to-image alignment.** We observe that EVALALIGN exhibits strong generalization capability.

| Method | T2I models | Object | Count | Color | Style | Spatial | Action | MAE |
|--------|-----------|--------|-------|-------|-------|---------|--------|-----|
| Human | - | 1.7373 | 1.3131 | 2.0000 | 1.9333 | 1.5952 | 1.8837 | 0 |
| EVALALIGN | 8 | 1.7203 | 1.3232 | 1.9565 | 1.9333 | 1.6547 | 1.8605 | 0.0256 |
| | 4 | 1.7832 | 1.3526 | 1.9637 | 1.9876 | 1.6891 | 1.8954 | 0.0469 |

## E.2    INSTRUCTION ENHANCEMENT EXPERIMENTS

Providing more contextual information for instructions enhances the performance of MLLMs. To further improve MLLM evaluation performance, we enhanced the prompts for both SFT and inference stages. As shown in Table 4, our experiments demonstrate that the enhanced prompts significantly increase evaluation accuracy. Specifically, the evaluation using enhanced instructions reduced the MAE metric by half, from 0.120 to 0.006, compared to the original instructions. Additionally, this approach consistently improved evaluation performance across different text-to-image models.

## E.3    MULIT-SCALING RESOLUTIONS EXPERIMENTS

In the design of LLaVA-Next, using multi-scale resolution images as input helps address the issue of detail information loss, which significantly impacts the evaluation of image faithfulness, such as assessing deformations in hands and faces. We conducted a multi-scale image training comparison experiment to validate this approach. The baseline was the 13B LLaVA model with $336\times336$ resolution input, while the comparison model used images at three resolutions ($336\times336$, $672\times672$, $1008\times1008$) as input. As shown in Table 5, training with multi-scale inputs significantly enhanced the model's understanding of image and achieved better evaluation performance.

# F    QUALITATIVE ANALYSIS

As shown in Figure 2, we present a comparison of different evaluation metrics on images generated by four models, including human annotated scores, EVALALIGN, ImageReward (Xu et al., 2024), HPSv2 (Wu et al., 2023), and PickScore (Kirstain et al., 2024). The digits in the figure represent the ranking for each evaluation metric, with darker colors indicating higher rankings. From the figure, it is evident that our proposed EvalAlign metric closely matches the human rankings across two

Table 4: **Ablation study on the enhancement of instructions.** Results are reported on image faithfulness under different instructions. We observe that enhanced instructions can significantly improves the evaluation metrics. MAE: mean absolute error.

| Method | Instruction | SDXL | Pixart | Wuerstchen | SDXL-Turbo | IF | SD v1.5 | SD v2.1 | LCM | MAE |
|---|---|---|---|---|---|---|---|---|---|---|
| Human | – | 2.1044 | 1.8606 | 1.7839 | 1.3854 | 1.3822 | 1.3818 | 1.1766 | 1.0066 | 0 |
| EVALALIGN | ✗ | 1.9565 | 1.9286 | 1.8565 | 1.1818 | 1.3419 | 1.4801 | 1.4078 | 1.1051 | 0.1201 |
| EVALALIGN | ✓ | 2.0443 | 1.9199 | 1.8012 | 1.3353 | 1.2960 | 1.4702 | 1.3221 | 1.0305 | 0.0064 |

Table 5: **Ablation study on multi-scale input.** Results are reported on image faithfulness under different input strategy. We observe that input with multi-scale resolution images can improves the evaluation metrics. MAE: mean absolute error.

| Method | Multi Scale | SDXL | Pixart | Wuerstchen | SDXL-Turbo | IF | SD v1.5 | SD v2.1 | LCM | MAE |
|---|---|---|---|---|---|---|---|---|---|---|
| Human | – | 2.1044 | 1.8606 | 1.7839 | 1.3854 | 1.3822 | 1.3818 | 1.1766 | 1.0066 | 0 |
| EVALALIGN | | 1.8105 | 1.9238 | 1.9325 | 1.2078 | 1.2247 | 1.4540 | 1.3012 | 1.0554 | 0.1358 |
| EVALALIGN | ✓ | 2.0443 | 1.9199 | 1.8012 | 1.3353 | 1.296 | 1.4702 | 1.3221 | 1.0305 | 0.0064 |

evaluation dimensions, demonstrating excellent consistency. Specifically, the numbers in the figure represent EVALALIGN scores for the corresponding evaluation aspect, with darker colors indicating higher scores and better generation performance. Note that if the text prompt does not specify a particular style, the style consistency score defaults to 0. From these results, it is evident that the same text-to-image model exhibits significant performance variation across different evaluation aspects.

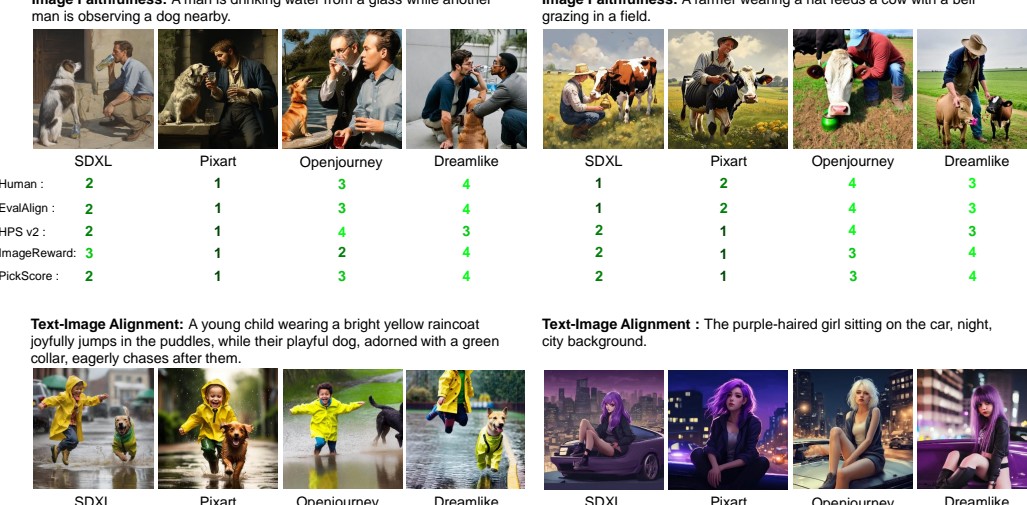

Figure 2: **Qualitative results of** EVALALIGN **dataset and benchmark.** As can be concluded, EVALALIGN is consistently aligned with fine-grained human preference in terms of image faithfulness and text-image alignment, while other methods fail to do so.