# OpenReview forum: "EvalAlign: Supervised Fine-Tuning Multimodal LLMs with Human-Aligned Data for Evaluating Text-to-Image Models"
_ICLR.cc/2025/Conference — Submitted to ICLR 2025_

### Official Review · Reviewer_rw7T · 2024-11-03

**Soundness:** 3
**Presentation:** 2
**Contribution:** 3
**Rating:** 5
**Confidence:** 4

**Summary:**

The paper introduces a novel method and dataset aimed at evaluating the quality of generated images, with a specific focus on image faithfulness and text-image alignment. The dataset was collected using detailed human feedback in a question-answer format, aiming to provide fine-grained insights into image quality. This dataset is then used to train a MLLM with SFT to evaluate generated images effectively. The proposed method is tested on the new dataset and compared against existing approaches, with results indicating its superior performance in terms of image faithfulness and text-image alignment.

**Strengths:**

- The paper introduces a dataset that includes explicit question-answer feedback, which could facilitate more detailed evaluation of generated images.
- Using MLLM with SFT for image evaluation is an interesting approach that could potentially enhance interpretability in assessing generated image quality.

**Weaknesses:**

## 1. Lack of Justification for Main Contributions
The paper's three key contributions are insufficiently supported by experimental validation and theoretical grounding:
   1. Although the dataset is described as having detailed human feedback covering "11 skills and 2 aspects," the experiments primarily focus on the 2 broad aspects. There is little exploration of the 11 specific skills, which would have been valuable given that the 2 aspects have been widely studied in prior works, such as [a].
   2. The method is claimed to enable "accurate, comprehensive, fine-grained, and interpretable" evaluations. However, the results mostly reflect the 2-aspect performance, with no evidence of superior fine-grained or interpretability-focused evaluation compared to previous methods.
   3. While the paper emphasizes cost-efficiency in terms of annotation and computation, this claim is questionable. The annotation process requires extensive human annotations, which is labor-intensive. Additionally, the method achieves optimal performance with a 34B MLLM model, which is computationally expensive.

## 2. Unclear Advantages Over Existing Datasets
- According to Table 1, the primary benefit of the proposed dataset seems to be its focus on the two-aspect evaluation. However, several prior datasets such as ImageReward, PickScore, and HPS(v2) implicitly address these aspects as well. While explicit question-based feedback is used in this work, it is not clear how this approach leads to better evaluation outcomes, especially since **a vast number of questions would likely be required to cover all image aspects comprehensively**.
- The paper does not adequately compare its approach with previous work like [a], which also includes detailed, multi-aspect human feedback via scoring rather than question-answering. The advantages of question-based feedback over scoring are not clearly demonstrated in terms of faithfulness or alignment evaluation.
- While the paper emphasizes cost-effectiveness, the dataset requires 130k annotations to achieve optimal results (Table 1). This annotation volume does not appear more economical than previous datasets.

## 3. Weak Experimental Results
- It is unclear whether models from other methods were trained on the proposed dataset to ensure a fair comparison, especially in Tables 2 and 3.
- The results in these tables do not consistently support the claims made. For instance, the 500 configuration does not show a clear optimal performance in Table 6, and there is no clear positive correlation between model size and performance improvements in Table 7.

## 4. Writing and Structure Issues
- Some sentences lack clarity and coherence. For instance, “the utilized synthesized images are treated as real images as they don’t explicitly recognize the problem of synthesized images with low image faithfulness” is confusing, especially since HPS(v2) aims to evaluate generated images.
- Writing structure should be improved. The key novel contributions of the method is unclear.
- There are some repeated sentences with similar meanings. It is better to re-write them to make the paper more concise.

## Conclusion

While the paper presents a promising approach with potential contributions in the form of a detailed dataset and a new evaluation method, it currently lacks sufficient support for its claims. The advantages over existing work remain unclear, and the experimental validation needs improvement. Therefore, the paper is not yet ready for acceptance in its current form.

[a] Rich Human Feedback for Text-to-Image Generation, CVPR 2024, best paper.

**Questions:**

please see weakness points above.

---

> ### Comment · Reviewer_rw7T · 2024-11-27
> **Final comments**
>
> Given that the authors have not offered a rebuttal and the majority of the other reviewers give negative ratings, I will remain my original score.

---

### Official Review · Reviewer_zeBD · 2024-11-03

**Soundness:** 3
**Presentation:** 3
**Contribution:** 3
**Rating:** 6
**Confidence:** 4

**Summary:**

In short, this paper presents EvalAlign, which collects a human-annotated preference dataset to fine-tune MLLMs to be evaluators for T2I generation. The paper has focused on two dimensions: (1) T2I alignment, (2) faithfulness, which is a well-accepted setting since AGIQA-3K (Li et al, 2023). Overall, the paper is technically sound, but I am a littble bit concerned on some methodology parts. Additionally, discussions for several pioneer works on T2I evaluation and MLLM as scorers are missing.

**Strengths:**

1. The dataset collection and annotation process is technically sound. The explicit prompting strategy (e.g. `Are there any issues with human face in the image, such as facial distortion, asymmetrical faces, abnormal facial features, unusual expressions in the eyes, etc?`) could be useful and scalable to better baseline MLLMs.
2. The evaluation part presents a benchmark on models, showing the high-correlation between the proposed scorer and human evaluation, which is good. It would become a useful metric.

**Weaknesses:**

I have some concerns on the paper.

First, I am a bit concerned on how the score is derived. From my current understanding, the final scores are derived from an average of the score outputs of several questions. Is ther `Human` column also obtained by so? If so, this might not be a good enough ground truth.

Second, well in Sec. 5.1 the author states that the test set images do no overlap with train set ones, they do come from the same 16 generation models. As the final evaluation only shows model-wise ranking consistency, this result might not enough exclude overfitting (e.g. memorizing on model specific styles, etc). I would encourage a further testing on several hold-out T2I generators.

Third, a minor question. Using SFT for LMM to score has been discussed by Q-Align (ICML2024), which finds out using logits are better than using `model.generate()` for scoring. It also has the ability for image faithfulness evaluation, please try to compare with it or discuss with it. Furthermore, for faithfulness evaluation (which is actually image quality, am I right?), the compared baselines are similarity-based metrics (which are, from their design, alignment-related metrics). I would suggest the authors to compare with some baselines related to T2I quality evaluation (inc. Q-Align) in this part.

**Questions:**

Please see weaknesses.

---

### Official Review · Reviewer_L57V · 2024-11-03

**Soundness:** 2
**Presentation:** 3
**Contribution:** 2
**Rating:** 3
**Confidence:** 5

**Summary:**

This paper proposes a method to evaluate the consistency of images and texts in the T2I generation process. Compared with other evaluation indicators such as ImageReward and HPS, it has higher consistency with human subjective preferences on 24 T2I models.

**Strengths:**

1. Using LLM to evaluate the quality of LLM is a very creative point. The author evaluates the T2I process through the I2T model, which is a new paradigm.
2. The experiment is relatively detailed, considering 24 generative models. Multiple dimensions are evaluated.
3. The illustrations are intuitive and beautiful, and Figure 1 reflects the central idea of ​​the article well.

**Weaknesses:**

1. The evaluation is done at the **model level**, not the **instance level**. This is a major flaw. In the actual evaluation process, the community is not only concerned with ranking the strength of the T2I model but whether each AIGC image is good enough. As far as I know, AIGC quality asessment [1,2] uses instance-level, because model-level evaluation is not very challenging (see Vbench [3] Figure 4, the correlation with subjective labels can easily reach 0.9). I hope the author can improve this point.
2. The author considered 24 models. Although the number is large, they are highly homogenized. For example, DeepFloyd IF uses three different conditions, which are considered three models, but they are the same. Including SD 1.4, 1.5, 2.0, and 2.1, the difference in visual effects is quite limited. However, for open-source models such as DALLE 3 and Midjourney, the dataset does not include them. From my subjective perspective, they are still slightly stronger than PixArt, and ignoring them will result in an incomplete dataset.

[1] Depicting Beyond Scores: Advancing Image Quality Assessment through Multi-modal Language Models
[2] Descriptive Image Quality Assessment in the Wild
[3] VBench: Comprehensive Benchmark Suite for Video Generative Models

**Questions:**

I would like to ask how long the author's evaluation takes. In my opinion, evaluation should be a task to assist generation. If the generated model is already large, using an estimator with 34B parameters will cost a lot, but only slightly improve the consistency with human subjective perception. I am not sure if it worth.
I am happy that the author analyzed the impact of different model sizes on performance, but the impact on time consumption also needs further explanation.

---

### Official Review · Reviewer_gtFg · 2024-11-04

**Soundness:** 2
**Presentation:** 2
**Contribution:** 2
**Rating:** 5
**Confidence:** 5

**Summary:**

This paper proposes EvalAlign, a metric characterized by accuracy, stability, and fine-grainedness. Evaluation on 24 text-to-image generation models shows that EvalAlign is more in line with human preferences than existing metrics and has certain application value in quality assessment.

**Strengths:**

1. This work has fine-grained annotation. Unlike previous datasets, EvalAlign annotates at three levels: animal faces, visibility of hands, and visibility of limbs. This detailed data enables the author to train an effective evaluation model.

2. The author promises open source code. And the experimental details are listed in the supplementary materials, which has strong reproducibility.

3. The writing of this article is quite fluent, and with appropriate illustrations, it is very easy for readers to understand.

**Weaknesses:**

1. The experimental part of this article has a big problem. Table 3 seems to have done a lot of experiments, but it is actually evaluated at the model level, not the instance level. This is not a challenging task, because everyone knows that PixArt draws well and SD 1.4 draws relatively poorly. Ranking the strengths of 24 models is far less meaningful than scoring a single image, that is, an end-to-end AIGC quality evaluation tool. In other words, which of the two images from the same model has higher quality is more important.

2. This paper only reviews coarse-grained datasets in related work, but does not consider fine-grained datasets. In addition, some AIGC-related dataset such as [1,2,3] was not considered. These datasets have fewer images but more annotations, and each image contains dozens of fine-grained annotations. Since fine-grained annotations are one of the major innovations of this paper, it is not comprehensive to only review coarse-grained datasets (i.e. only two or three annotations, or even less than one per images).

[1] PKU-AIGIQA-4K: A Perceptual Quality Assessment Database for Both Text-to-Image and Image-to-Image AI-Generated Images

[2] PKU-AIGI-500K: A Neural Compression Benchmark And Model for AI-Generated Images

[3] PKU-I2IQA: An image-to-image quality assessment database for ai generated images

**Questions:**

I am very concerned about Table 8 of this paper. Are the calculated KRCC and PLCC based on the instance level? If it is at the model level, it is recommended that the author modify it according to the content of weakness. If it is indeed at the instance level, I hope the author will focus on the analysis at this step, which is more important than the scores of each model listed in Table 3. Also, the authors can check the Weaknesses, and address them point-by-point in the response, which would be helpful.

---

> ### Comment · Reviewer_gtFg · 2024-12-02
>
> Since the authors didn't provide the response to my concerns, I decide to keep my original rating.

---

### Meta-Review · Area_Chair_aY85 · 2024-12-19

**Metareview:**

This paper collects a human-annotated preference dataset to fine-tune MLLMs as evaluators for T2I models.

The strength of the paper is: 1) a fine-grained evaluation data sets 2) Experiments on large number (24) generative models

However, the major weaknesses are: 1) the evaluation is done on model level, not image level 2) related works are missing or not compared like the ones with fine-grained or rich human feedback 3)  more detailed human feedback are provided, but the paper focused on two major dimensions (T2I alignment, faithfulness) are not

**Additional Comments On Reviewer Discussion:**

No rebuttal are provided by authors.

---

### Decision · Program_Chairs · 2025-01-22

Reject